# Systematic Review and Meta-Analysis of Clinical Efficacy and Safety of Meropenem-Vaborbactam versus Best-Available Therapy in Patients with Carbapenem-Resistant Enterobacteriaceae Infections

**DOI:** 10.3390/ijms25179574

**Published:** 2024-09-04

**Authors:** Alexandra Bucataru, Adina Turcu-Stiolica, Daniela Calina, Andrei Theodor Balasoiu, Ovidiu Mircea Zlatian, Andrei Osman, Maria Balasoiu, Alice Elena Ghenea

**Affiliations:** 1Medical Doctoral School, University of Medicine and Pharmacy of Craiova, 200349 Craiova, Romania; alexandra.catana95@gmail.com; 2Infectious Disease Department, Victor Babes University Hospital Craiova, 200515 Craiova, Romania; 3Pharmacoeconomics Department, University of Medicine and Pharmacy of Craiova, 200349 Craiova, Romania; 4Department of Clinical Pharmacy, University of Medicine and Pharmacy of Craiova, 200349 Craiova, Romania; calinadaniela@gmail.com; 5Department of Ophthalmology, University of Medicine and Pharmacy of Craiova, 200349 Craiova, Romania; andrei_theo@yahoo.com; 6Department of Bacteriology-Virology-Parasitology, University of Medicine and Pharmacy of Craiova, 200349 Craiova, Romania; ovidiu.zlatian@umfcv.ro (O.M.Z.); maria.balasoiu@umfcv.ro (M.B.); gaman_alice@yahoo.com (A.E.G.); 7Department of Anatomy and Embryology, Faculty of Dental Medicine, University of Medicine and Pharmacy of Craiova, 200349 Craiova, Romania; andrei.osman@umfcv.ro; 8Department ENT & Clinical Emergency County Hospital of Craiova, 200642 Craiova, Romania

**Keywords:** meropenem-vaborbactam, carbapenem-resistant Enterobacteriaceae, adverse events

## Abstract

Antimicrobial resistance is increasingly concerning, causing millions of deaths and a high cost burden. Given that carbapenemase-producing Enterobacterales are particularly concerning due to their ability to develop structural modifications and produce antibiotic-degrading enzymes, leading to high resistance levels, we sought to summarize the available data on the efficacy and safety regarding the combination of meropenem-vaborbactam (MV) versus the best available therapy (BAT). Articles related to our objective were searched in the PubMed and Scopus databases inception to July 2024. To assess the quality of the studies, we used the Cochrane risk-of-bias tool, RoB2. The outcomes were pooled as a risk ratio (RR) and a 95% confidence interval (95%CI). A total of four published studies were involved: one retrospective cohort study and three phase 3 trials, including 432 patients treated with MV and 426 patients treated with BAT (mono/combination therapy with polymyxins, carbapenems, aminoglycosides, colistin, and tigecycline; or ceftazidime-avibactam; or piperacillin-tazobactam). No significant difference in the clinical response rate was observed between MV and the comparators at the TOC (RR = 1.29, 95%CI [0.92, 1.80], *p* = 0.14) and EOT (RR = 1.66, 95%CI [0.58, 4.76], *p* = 0.34) visits. MV was associated with a similar microbiological response as the comparators at TOC (RR = 1.63, 95%CI [0.85, 3.11], *p* = 0.14) and EOT assessment (RR = 1.16, 95%CI [0.88, 1.54], *p* = 0.14). In the pooled analysis of the four studies, 28-day all-cause mortality was lower for MV than the control groups (RR = 0.47, 95%CI [0.24, 0.92], *p* = 0.03). MV was associated with a similar risk of adverse events (AEs) as comparators (RR = 0.79, 95%CI [0.53, 1.17], *p* = 0.23). Additionally, MV was associated with fewer renal-related AEs than the comparators (RR = 0.32, 95%CI [0.15, 0.66], *p* = 0.002). MV was associated with a similar risk of treatment discontinuation due to AEs (RR = 0.76, 95%CI [0.38, 1.49], *p* = 0.42) or drug-related AEs (RR = 0.56, 95%CI [0.28, 1.10], *p* = 0.09) as the comparators. In conclusion, MV presents a promising therapeutic option for treating CRE infections, demonstrating similar clinical and microbiological responses as other comparators, with potential advantages in mortality outcomes and renal-related AEs.

## 1. Introduction

The impact of antimicrobial resistance (AMR) is increasingly concerning, with estimates from 2019 indicating that AMR was responsible for approximately 1.3 million deaths worldwide [1]. Projections suggest that without effective interventions, the annual death number could rise to 10 million by 2050 [2]. The economic burden of AMR is also substantial. In Europe, AMR is associated with costs exceeding nine billion euros annually [3]. At the same time, in the United States, the Centers for Disease Control and Prevention (CDC) estimates that AMR incurs an additional $20 billion in direct healthcare expenses and $35 billion in lost productivity each year [4].

The severity of antibiotic resistance is escalating, particularly with the prevalence of multi-drug-resistant (MDR) bacteria and the development of resistance mechanisms against many antibiotics [5]. Enterobacterales primarily develop resistance to carbapenems through the production of carbapenemase enzymes, with additional mechanisms involving efflux pump activity and porin modification [6].

The most clinically significant carbapenemases include *Klebsiella pneumoniae* carbapenemase (KPC), New Delhi metallo-β-lactamase (NDM), Verona integron-encoded metallo-β-lactamase (VIM), Imipenemase metallo-β-lactamase (IMP), and Oxacillinase-48 (OXA-48) [7]. The distribution varies geographically, with KPC being most common in the United States and Europe, while NDM is more prevalent in South Asia [8].

Carbapenemase-producing Enterobacterales (CRE) are particularly concerning due to their ability to develop structural modifications and produce antibiotic-degrading enzymes, leading to high resistance levels [9]. In response to this critical issue, novel treatment alternatives have been developed. One effective strategy combines beta-lactam with beta-lactamase inhibitors, such as tazobactam, sulbactam, and clavulanic acid, to combat beta-lactamase-mediated resistance [10]. Additionally, new combinations like ceftazidime-avibactam, ceftolozane-tazobactam, and meropenem-vaborbactam (MV) have enhanced the efficacy of beta-lactam antibiotics against Gram-negative infections [11].

According to the Infectious Diseases Society of America (IDSA) guidelines, ceftazidime-avibactam or meropenem-vaborbactam are recommended treatments, particularly for KPC-producing organisms. In cases where resistance is present, alternatives like cefiderocol or a combination of polymyxins, tigecycline, and fosfomycin are suggested, though these may have higher toxicity or lower efficacy [12,13].

Meropenem is a bactericidal agent that targets penicillin-binding proteins (PBPs) in the bacterial cell wall, inhibiting the formation of peptidoglycan cross-links necessary for cell wall synthesis, ultimately leading to bacterial cell death [14]. Vaborbactam, a novel beta-lactamase inhibitor with a cyclic boronic acid structure, is specifically designed to target KPC-type carbapenemases as well as other class A and C β-lactamases [15]. It enters bacterial cells via OmpK35 and OmpK36 porins and acylates the active sites of these enzymes, forming a covalent adduct that inhibits the enzyme’s activity by mimicking the tetrahedral transition state of the enzyme’s normal hydrolysis process [14,16]. While vaborbactam alone does not possess antibacterial activity, it is used in combination with meropenem to inhibit the degradation of meropenem by serine beta-lactamases, thereby enhancing the efficacy of the antibiotic [15].

MV has demonstrated excellent in vitro activity against CRE isolates, including those resistant to ceftazidime-avibactam. This combination showed greater activity compared to meropenem alone and most other comparator drugs, particularly against CRE and KPC-producing organisms [17,18,19].

To the best of our knowledge, no systematic review and meta-analysis has been conducted on the efficacy and safety of the MV combination compared with the best available therapy (BAT) for the treatment of CRE infections. Our study results will contribute to informing future practices aimed at optimizing the treatment of CRE infections.

## 2. Materials and Methods

### 2.1. Study Search

We have conducted this systematic review and meta-analysis to estimate the pooled outcomes of MV combination compared with the BAT for the treatment of Carbapenem-Resistant Enterobacteriaceae Infections. The study protocol was registered in the PROSPERO database (registration no. CRD42024566856) to ensure transparency in the systematic review process and to minimize the risk of duplication of research. The results of the review were reported using the Preferred Reporting Items for Systematic Reviews and Meta-Analysis (PRISMA) 2020 checklist (Appendix A).

We searched the PubMed and Scopus databases as well as the clinical trials registries of ClinicalTrials.gov from inception to July 2024. The following search terms were used: (vabomere OR ((meropenem OR RPX-2014) AND (vaborbactam OR RPX-7009)) AND (“carbapenem-resistant Enterobacteriaceae”). The reference lists of relevant articles were also examined for additional eligible articles.

### 2.2. Inclusion and Exclusion Criteria

We included studies involving patients with Carbapenem-Resistant Enterobacteriaceae Infections treated with the MV combination. The studies must measure the efficacy and safety of MV (MV group) in comparison with BAT (other antibiotics, C group). Studies were excluded if they met the following criteria: (1) in vitro activity research; (2) animal studies; (3) pharmacokinetic/pharmacodynamic assessments; (4) no measurements of the primary and secondary outcomes for both groups of patients (MV group and C group).

The primary outcome was a clinical cure at the end of treatment (EOT). Secondary outcomes included: (1) clinical cure at the test of cure (TOC), (2) microbiological cure at EOT, (3) microbiological cure at TOC, (4) 28-day all-cause mortality, (5) adverse events, (6) renal-related adverse events; (7) discontinuation of study drug due to adverse events, and (8) discontinuation of study drug due to drug-related AE.

### 2.3. Study Selection and Data Extraction

The articles found through the electronic database searches were exported to a Microsoft Excel spreadsheet, where duplicate studies were removed. Two authors (A.B. and O.M.Z.) independently screened and reviewed each study. Disagreements between the reviewers were resolved by a third reviewer (A.T.-S.). The reasons for excluding the articles were documented at each step.

To assess the quality of the studies, we used the Cochrane risk-of-bias tool, RoB2. Two authors (A.B. and O.M.Z.) independently evaluated the methodological quality of each study. Disagreements between the reviewers were resolved by a third reviewer (A.T.-S.). The following components were evaluated: (Q1) Random sequence generation (selection bias); (Q2) Allocation concealment (selection bias); (Q3) Blinding of participants and personnel (performance bias); (Q4) Blinding of outcomes assessment (detection bias); (Q5) Incomplete outcome data (attrition bias); and (Q6) Selective reporting (reporting bias); (Q7) Other bias.

We collected the first author’s last name, year of publication, sample characteristics, study design, study setting, response rates, and outcome measures. Missing data were addressed by trying to reach the corresponding authors of the studies.

### 2.4. The Articles Selection Scheme

Initially, 619 records were identified from PubMed (n = 270) and Scopus (n = 349). After excluding two duplicate records and 600 irrelevant articles by title and abstract, 17 reports were reviewed by eligibility. Further exclusion of 13 studies for various reasons (e.g., pediatric population) resulted in 4 studies [20,21,22,23] that met the selection criteria. This systematic review followed the Preferred Reporting Items for Systematic Reviews and Meta-Analyses (PRISMA) guidelines. The algorithm for study selection is shown in Figure 1.

A total of four published studies were involved, as shown in Appendix A. These four studies included one retrospective cohort study [21] and three phase 3 trials [20,22,23], including 432 patients treated with MV and 426 patients treated with BAT (mono/combination therapy with polymyxins, carbapenems, aminoglycosides, colistin, tigecycline; or ceftazidime-avibactam; or piperacillin-tazobactam). All of them were multicenter and multinational studies that focused on adult patients (Appendix A). One study enrolled patients with bacteremia from common sources such as urinary tract, intra-abdominal, and respiratory tract [21], while another investigated a population with cUTI/AP [22]. The third and fourth studies assessed patients with serious bacterial infections, including cUTI, AP, HABP/VABP, and cIAI [20,23]. Overall, females accounted for a significant proportion of the population in both groups, 63.73% and 58.08%, respectively (Appendix A). The most common type of site infection was the urinary tract (cUTI/AP), both in the MV and control groups (Appendix A). Appendix A summarizes the microbiological distribution. *Escherichia coli* was the most frequent pathogen in both groups (37.6%, respectively, 31.3%), followed by *Klebsiella pneumoniae* (27.1%, respectively 31.3%).

Except for the risk of selection and performance biases owing to the open-label design of the study by Ackley et al. [21], the other studies had a low risk of bias across all seven fields, as shown in Figure 2.

### 2.5. Statistical Analysis

The extracted data were imported from Excel into R statistical software (R-4.4.1. for Windows). A random-effects model or fixed-effects model was fitted to the data according to the amount of heterogeneity among the included studies. The amount of heterogeneity was estimated by calculating tau^2^ using the DerSimonian–Laird estimator, performing the Q-test for heterogeneity and reporting Higgins and Thompson’s I^2^ statistic. If any heterogeneity was detected (tau^2^ > 0, *p*-value < 0.05), a random-effects model, Mantel–Haenszel, was performed. The heterogeneity of the results was visually examined via forest plots with pooled estimates. The outcomes were pooled as a risk ratio (RR) and a 95% confidence interval (95%CI).

The rank correlation test (Kendall’s tau) and the regression test were used to check for funnel plot asymmetry, with a *p*-value < 0.05 indicating no asymmetry and no publication bias in the included studies.

## 3. Results

### 3.1. Microbiological Characteristics of the Studies

*Klebsiella pneumoniae* and *Escherichia coli* were the primary causative CRE organisms in both groups. They were more commonly isolated in the MV group, but this difference was not significant, as shown in Figure 3. *Proteus mirabilis* was significantly more isolated in the C group than in the MV group (RR = 0.37, 95%CI 0.16 to 0.85, *p* = 0.02).

### 3.2. Clinical Cure at the End of Treatment

A total of three studies were included in the analysis, comprising 273 patients treated with MV and 317 patients treated with the comparators. The estimated pooled risk ratio is 1.66 (95%CI, 0.58 to 4.76) after fitting the data with a random-effects model. We found high heterogeneity (tau^2^ = 0.77, I^2^ = 91%, χ^2^ = 23.11, *p*-value < 0.00001). Therefore, as shown in Figure 4A, no significant differences were observed between the clinical cure at the EOT with MV and the clinical cure at the end of treatment with C (Z = 0.95, *p*-value = 0.34). Neither the rank correlation (Kendall’s tau = −0.333, *p*-value = 0.75) nor the regression test (Z = −0.858, *p* = 0.391) indicated any funnel plot asymmetry, as shown in Figure 4B.

### 3.3. Clinical Cure at the Test of Cure

No significant difference in the clinical cure at the TOC visit was observed between MV and the comparators (RR = 1.29, 95%CI 0.92 to 1.80, *p* = 0.14). Four studies, including 273 patients treated with MV and 317 treated with comparators, were included in this meta-analysis. A random-effects model was used to pool the data, as shown in Figure 5A. High heterogeneity was observed between the studies (tau^2^ = 0.06, I^2^ = 68%, χ^2^ = 9.48, *p*-value = 0.02), and the data were fitted using a random-effects model. Neither the rank correlation (Kendall’s tau = 1.0, *p*-value = 0.083) nor the regression test (Z = 1.901, *p* = 0.057) indicated any funnel plot asymmetry, as shown in Figure 5B.

### 3.4. Microbiological Eradication Rate at the End of Treatment

A total of four studies were included in the analysis, comprising 256 patients in the MV group and 256 in the C group. The estimated pooled RR based on the random-effects model was 1.16 (95%CI, [0.88, 1.54]). According to the Q-test, there was a high amount of heterogeneity (tau^2^ = 0.05, I^2^ = 73%, χ^2^ = 11.10, *p* = 0.01), as shown in Figure 6A. No significant difference in the microbiological eradication rate at EOT was observed between the MV and C groups (Z = 1.47, *p* = 0.14). Neither the rank correlation (Kendall’s tau = −0.333, *p*-value = 0.750) nor the regression test (Z = −1.711, *p* = 0.087) indicated any funnel plot asymmetry, as shown in Figure 6B.

### 3.5. Microbiological Eradication Rate at Test of Cure

A total of three studies were included in the analysis, comprising 247 patients in the MV group and 220 in the C group. The estimated pooled RR based on the random-effects model was 1.63 (95%CI, [0.85, 3.11]). According to the Q-test, there was a high amount of heterogeneity (tau^2^ = 0.23, I^2^ = 73%, χ^2^ = 7.31, *p* = 0.03), as shown in Figure 7A. No significant difference in the microbiological eradication rate at TOC was observed between the MV and C groups (Z = 1.47, *p* = 0.14). Neither the rank correlation (Kendall’s tau = 1.0, *p*-value = 0.333) nor the regression test (Z = 1.401, *p* = 0.161) indicated any funnel plot asymmetry, as shown in Figure 7B.

### 3.6. Day-28 All-Cause Mortality

A total of four studies were included in the analysis, including 353 patients in the MV group and 408 in the C group. The estimated pooled RR based on the fixed-effects model was 0.47 (95%CI, [0.24, 0.92]). According to the Q-test, there was no significant amount of heterogeneity (I^2^ = 0%, χ^2^ = 2.28, *p* = 0.52), as shown in Figure 8A. In the pooled analysis, 28-day all-cause mortality was significantly higher for the C group than for the MV group (Z = 2.20, *p* = 0.03), as shown in Figure 8A. Neither the rank correlation (Kendall’s tau = −0.333, *p*-value = 0.750) nor the regression test (Z = −0.687, *p* = 0.492) indicated any funnel plot asymmetry, as shown in Figure 8B.

### 3.7. Risk of Adverse Events of Meropenem-Vaborbactam and Comparators

Overall, fewer adverse events were observed in the MV group than in the comparator group, but this difference was not statistically significant (Z = 1.20, *p* = 0.23). A total of four studies were included in the analysis, comprising 380 patients in the MV group and 426 in the C group. The estimated pooled RR based on the random-effects model was 0.79 (95%CI, [0.53, 1.17]). According to the Q-test, there was a significant amount of heterogeneity (tau^2^ = 0.08, I^2^ = 53%, χ^2^ = 6.37, *p* = 0.10), as shown in Figure 9A. The rank correlation (Kendall’s tau = −0.667, *p*-value = 0.333) did not indicate funnel plot asymmetry, but nor did the regression test (Z = −2.45, *p* = 0.014), as shown in Figure 9B.

### 3.8. Risk of Renal-Related Adverse Events between Meropenem-Vaborbactam vs. Comparators

The renal-related AEs were acute renal failure, renal impairment, or renal failure. MV was associated with fewer renal-related AEs than comparators (Z = 3.10, *p* = 0.002, RR = 0.32, 95%CI 0.15 to 0.66). No amount of heterogeneity was found among the four included studies (tau^2^ = 0, I^2^ = 0%, χ^2^ = 1.41, *p* = 0.70), as shown in Figure 10A, and a fixed-effects model was used for fitting the data. Neither the rank correlation (Kendall’s tau = 0.0, *p*-value = 1.0) nor the regression test (Z = −0.241, *p* = 0.810) indicated any funnel plot asymmetry, as shown in Figure 10B.

### 3.9. Discontinuation of Study Due to Drug-Related Adverse Events

Drug-related AEs occurring more in MV-treated patients included diarrhea, anemia, and hypokalemia. Drug-related AEs occurring more in BAT-treated patients included sepsis, septic shock, diarrhea, anemia, hypotension, and acute renal failure. MV was associated with a similar risk of treatment discontinuation due to AEs; participants exited the study entirely due to AEs as the comparators (*p* = 0.42). This means they were no longer involved in any aspect of the study, including follow-ups, assessments, or data collection in the MV and C groups. A total of four studies were included in the analysis, comprising 380 patients in the MV group and 426 in the C group. The estimated pooled RR based on the random-effects model was 0.76 (95%CI, [0.38, 1.49]). According to the Q-test, there was no amount of heterogeneity (tau^2^ = 0, I^2^ = 0%, χ^2^ = 0.33, *p* = 0.95), as shown in Figure 11A. Neither the rank correlation (Kendall’s tau = 0.0, *p*-value = 1.0) nor the regression test (Z = 0.096, *p* = 0.924) indicated any funnel plot asymmetry, as shown in Figure 11B.

### 3.10. Discontinuation of Study Drug Due to Drug-Related AEs

MV was associated with a similar risk of treatment discontinuation due to drug-related AEs as comparators (*p* = 0.09). The participants stopped taking the drug being tested but many continued in the study for follow-up or other assessments, both in the MV and C groups. The estimated pooled RR based on the random-effects model was 0.56 (95%CI, [0.28, 1.10]). A total of three studies were included in the analysis, comprising 354 patients in the MV group and 321 in the C group. According to the Q-test, there was no amount of heterogeneity (tau^2^ = 0, I^2^ = 0%, χ^2^ = 0.42, *p* = 0.81), as shown in Figure 12A. Neither the rank correlation (Kendall’s tau = 0.333, *p*-value = 0.064) nor the regression test (Z = 0.198, *p* = 0.843) indicated any funnel plot asymmetry, as shown in Figure 12B.

## 4. Discussion

In this meta-analysis, four RCTs were reviewed to compare MV with other antibiotic regimens in terms of efficacy and safety for the treatment of CRE infections. The comprehensive study supports that MV achieves clinical results that are at least comparable to those of the comparators.

The effectiveness of MV in the treatment of CRE infections was similar to that of the comparators in terms of clinical efficacy and microbiological response. Clinical efficacy was demonstrated at both the EOT and TOC assessments. For example, the Ackley et al. [21] study reported an EOT clinical cure rate of 69.2% for MV compared to 61.9% for BAT. Similarly, the Kaye et al. [22] trial documented a TOC clinical cure rate of 90.6% for MV versus 86.3% for BAT. These findings affirm the consistent efficacy of MV across diverse studies and infection types, including cUTI and APN. Supporting these findings, Zhang et al. [24] and Shields et al. [25] also demonstrated a favorable clinical response in the treatment of CRE infections.

The microbiological efficacy of MV was particularly notable. The Basetti 2019 study highlighted a TOC microbiological eradication rate of 69.6% compared to just 26.7% for BAT [18]. Previous in vitro studies have supported the favorable microbiological response of MV [18,26,27].

Interesting conclusions were provided by the analysis of 28-day all-cause mortality rates, which indicates that MV is associated with significantly lower 28-day mortality compared to BAT in patients with Carbapenem-resistant Enterobacteriaceae infections. In another meta-analysis comparing cefiderocol with the best available comparators in carbapenem-resistant Gram-negative bacteria, all-cause mortality at 28 days did not differ among them [28]. Several studies in the literature have demonstrated that the early administration of MV, particularly within the first 48 h of infection, is associated with a substantial decrease in mortality rates, suggesting that timely intervention can be crucial in improving patient survival outcomes. This evidence emphasizes the essential role of early MV administration in managing CRE infections [29,30].

MV demonstrated a safety profile comparable to that of the comparators. The most common adverse events were gastrointestinal symptoms, which were generally manageable and did not lead to significant rates of treatment discontinuation. In their study, Rubino et al. [31] revealed that no participants discontinued the study due to adverse events, and no serious adverse events were observed, which aligns with the findings of our meta-analysis. Furthermore, MV was associated with a lower incidence of renal-related adverse events compared to the comparators, indicating a potentially favorable renal safety profile. These findings are corroborated by other studies [30,32], which reported that MV therapy was associated with a good safety profile and a low risk of nephrotoxicity.

This meta-analysis had several limitations. First, the number of studies and patients was relatively limited, necessitating further large-scale RCTs to confirm these findings. Second, the heterogeneity in study design and patient populations introduces variability that may affect the generalizability of the results. Third, the relatively small sample sizes in some studies, particularly for specific infection types, may reduce the statistical power to detect differences in outcomes.

## 5. Conclusions

In conclusion, MV presents a promising therapeutic option for treating CRE infections, demonstrating similar clinical and microbiological responses as other comparators, with potential advantages in mortality outcomes and renal-related adverse events.

## Figures and Tables

**Figure 1 ijms-25-09574-f001:**
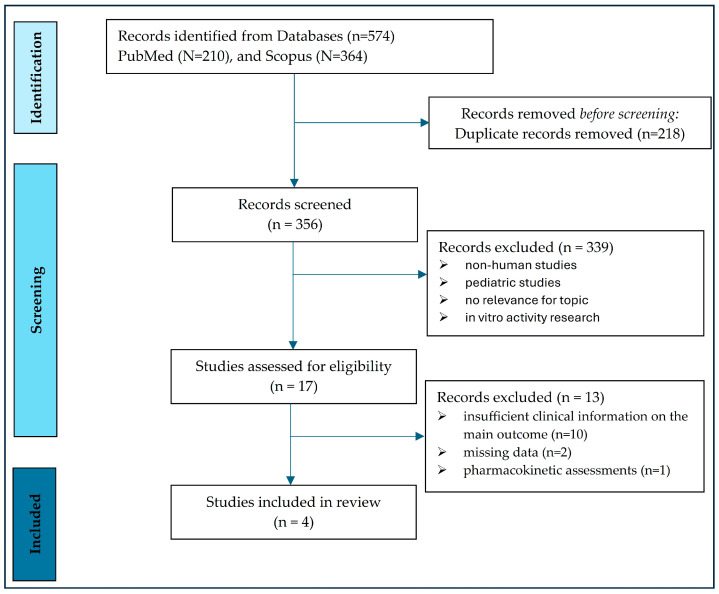
PRISMA flow chart of the study selection.

**Figure 2 ijms-25-09574-f002:**
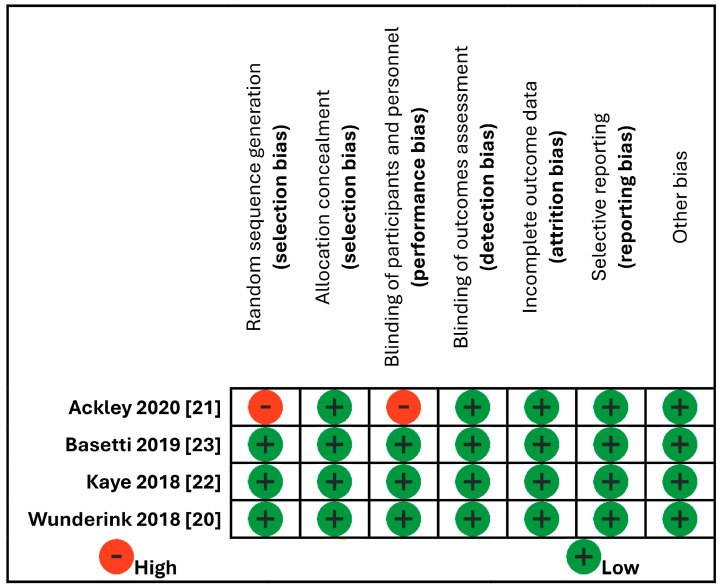
Summary of the risk of bias [20,21,22,23].

**Figure 3 ijms-25-09574-f003:**
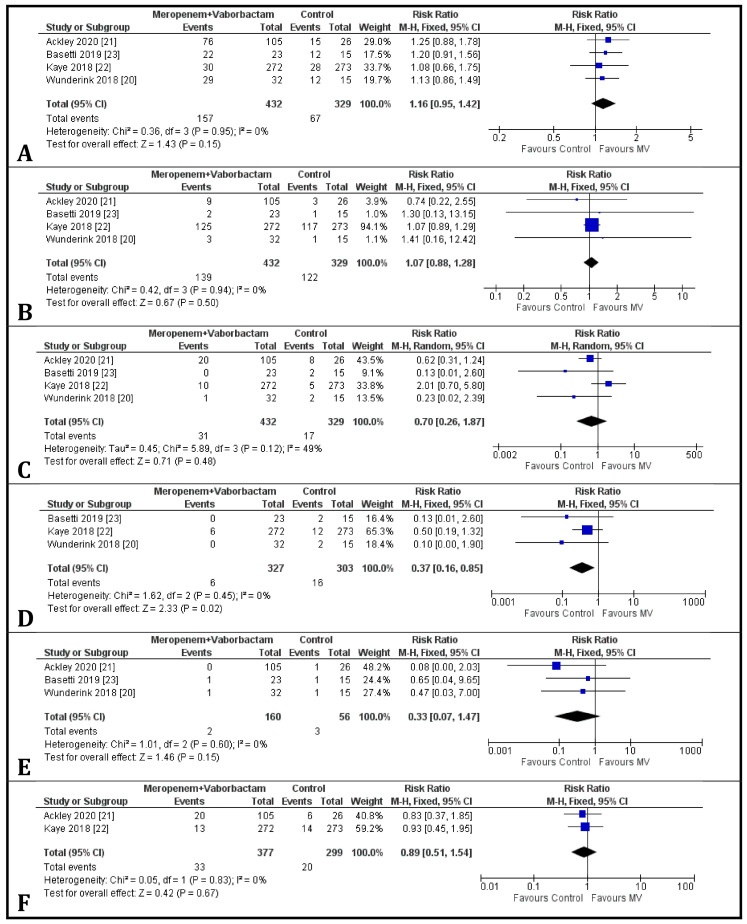
Baseline pathogens. (**A**) *Klebsiella pneumoniae*. (**B**) *Escherichia coli*. (**C**) *Enterobacter cloacae* sp. (**D**) *Proteus mirabilis*. (**E**) *Serratia marcescens*. (**F**) *Enterococcus faecalis* [20,21,22,23].

**Figure 4 ijms-25-09574-f004:**
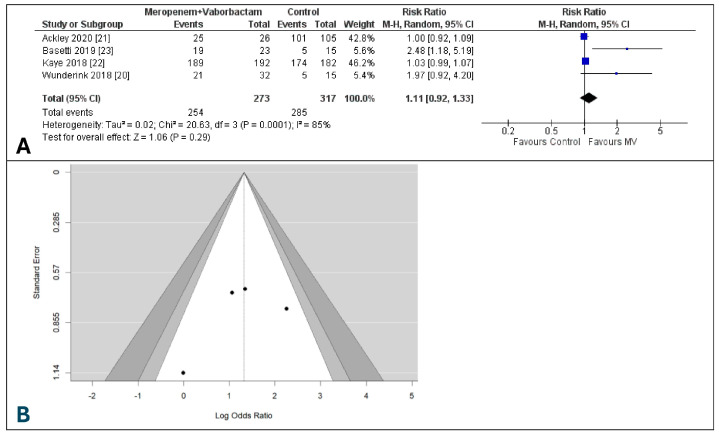
(**A**) Forest plot of the studies included in the meta-analysis of the clinical cure at the end of the treatment of meropenem-vaborbactam vs. comparators. (**B**) The funnel plot for the publication bias assessment of the included studies [20,21,22,23].

**Figure 5 ijms-25-09574-f005:**
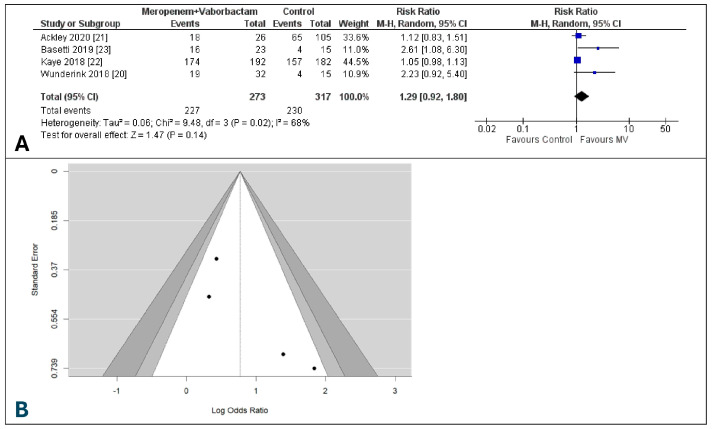
(**A**) Forest plot of the studies included in the meta-analysis of the clinical cure at the test of cure between meropenem-vaborbactam vs. comparators. (**B**) The funnel plot for the publication bias assessment of the included studies [20,21,22,23].

**Figure 6 ijms-25-09574-f006:**
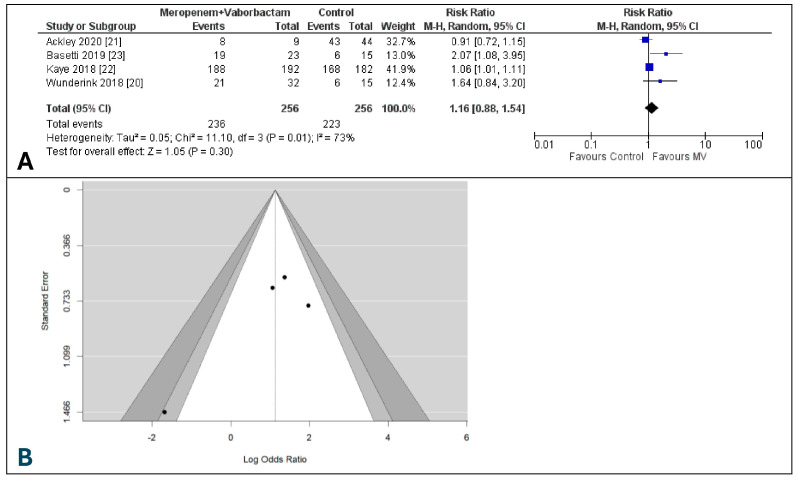
(**A**) Forest plot of the studies included in the meta-analysis of the microbiological rate at the end of treatment between meropenem-vaborbactam and comparators. (**B**) The funnel plot for the publication bias assessment of the included studies [20,21,22,23].

**Figure 7 ijms-25-09574-f007:**
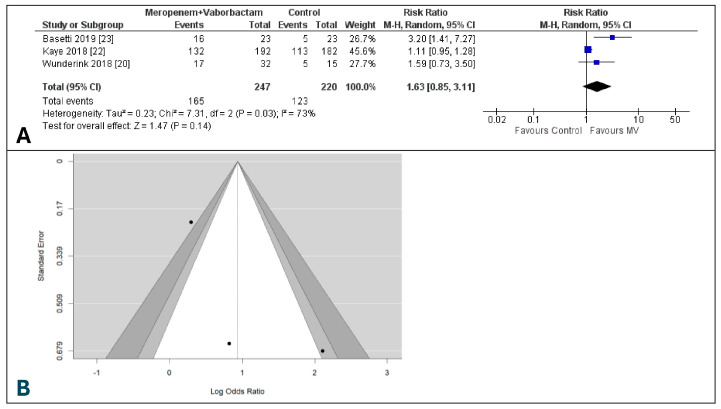
(**A**) Forest plot of the studies included in the meta-analysis of the microbiological rate at the test of cure between meropenem-vaborbactam and comparators. (**B**) The funnel plot for the publication bias assessment of the included studies [20,22,23].

**Figure 8 ijms-25-09574-f008:**
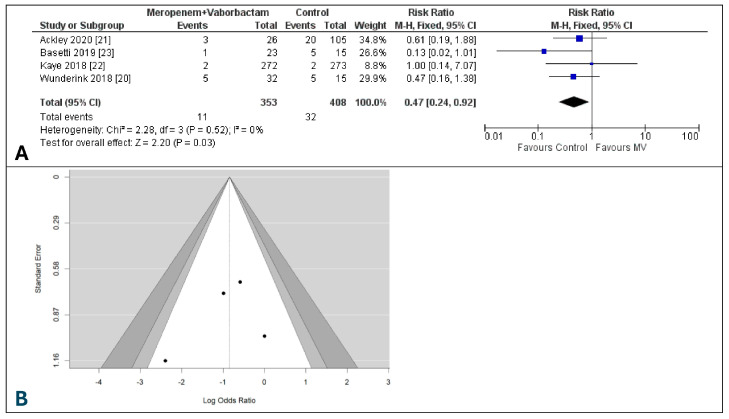
(**A**) Forest plot of the 28-day all-cause mortality between meropenem-vaborbactam and comparators. (**B**) The funnel plot for the publication bias assessment of the included studies [20,21,22,23].

**Figure 9 ijms-25-09574-f009:**
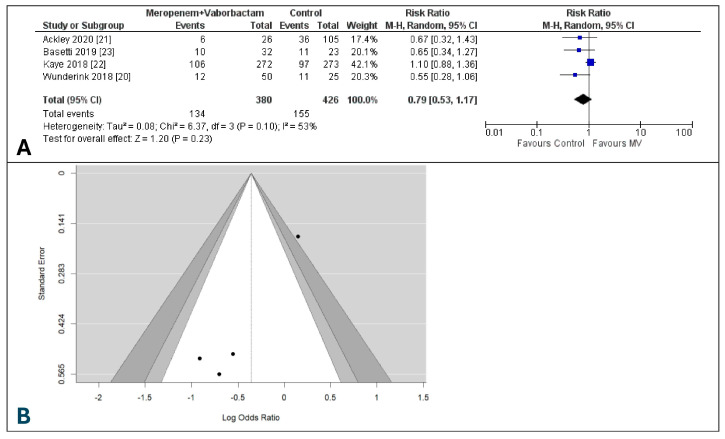
(**A**) Forest plot of the risk of adverse events between meropenem-vaborbactam and comparators. (**B**) The funnel plot for the publication bias assessment of the included studies [20,21,22,23].

**Figure 10 ijms-25-09574-f010:**
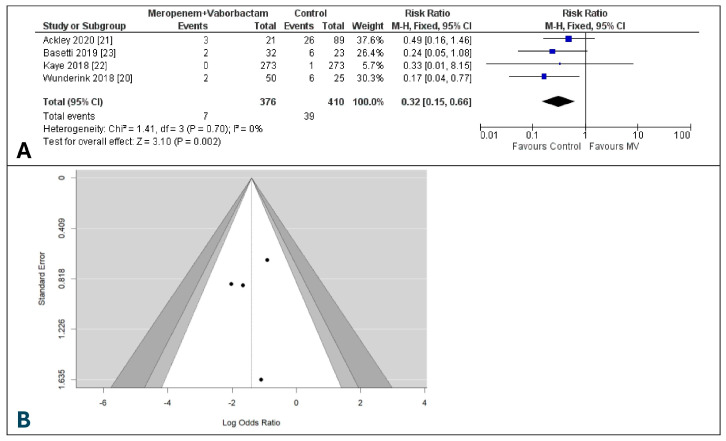
(**A**) Forest plot of the risk of renal-related adverse events between meropenem-vaborbactam and comparators. (**B**) The funnel plot for the publication bias assessment of the included studies [20,21,22,23].

**Figure 11 ijms-25-09574-f011:**
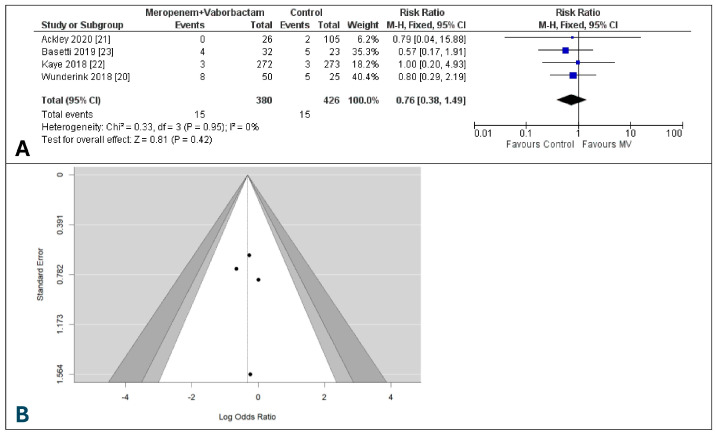
(**A**) Forest plot of risk of discontinuation of study due to drug-related adverse events between meropenem-vaborbactam and comparators. (**B**) The funnel plot for the publication bias assessment of the included studies [20,21,22,23].

**Figure 12 ijms-25-09574-f012:**
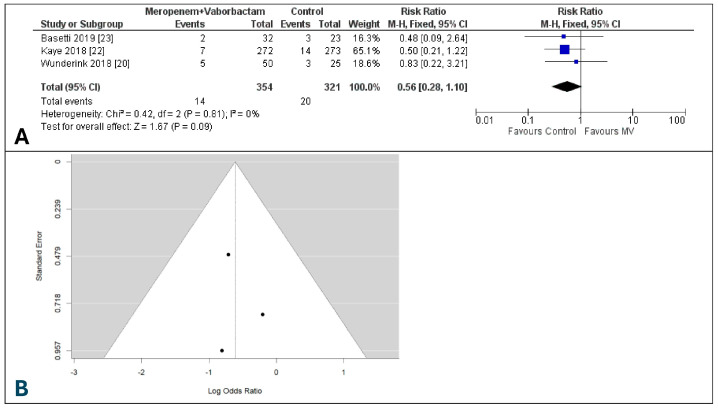
(**A**) Forest plot of risk of discontinuation of study drug due to drug-related adverse events between meropenem-vaborbactam and comparators. (**B**) The funnel plot for the publication bias assessment of the included studies [20,22,23].

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
