# Peer review of "Systematic Review and Meta-Analysis of Clinical Efficacy and Safety of Meropenem-Vaborbactam versus Best-Available Therapy in Patients with Carbapenem-Resistant Enterobacteriaceae Infections"

_ijms, 2024, doi:10.3390/ijms25179574_

Round 1

Reviewer 1 Report

Comments and Suggestions for Authors

The submitted manuscript “Systematic Review and Meta-analysis of Clinical Efficacy and Safety of Meropenem-Vaborbactam versus Best-Available Therapy in Patients with Carbapenem-resistant Enterobacteriaceae Infections” focus on an interesting and relevant subject regarding human health and antimicrobials resistance. Considering nowadays pertinence of the subject in the “one health” perspective, it is important to analyse scientific data and results to support the cautious use and efficacy of antimicrobials.

The manuscript aims to review and examine the efficacy and security of Meropenem-Vaborbactam versus other therapies. Even the number of patients of the different studies included in the analysis can be considered good, the number of studies elected in the Meta-analysis seems poor, maybe due to the scarce number of studies published in this subject or the exclusion criteria.

In general, the conceptualization of the study is correct.

However, the small number of studies elected to the analysis compromises the relevance of the present study and should be considered as a main limitation.

This kind of scientific study is important and brings useful information to the literature.

The Abstract is satisfactory and well written. It provides a correct information regarding the content of the manuscript (objectives, materials and methods, results and discussion) in a clear and simple language.

Line: Can the authors specify better what consider BAT (best available therapy)? To the readers, BAT gives no useful information…

The keywords are adequate to the content of the study.

The introduction provides correct information regarding AMR and its importance, but can be improved.

Lines 89-91: Can again clarify better what is considered the BAT (best available therapy). This concept is too unspecific to understand the usefulness of this manuscript.

Lines 91-92: This sentence seems too ambitious to be in the introduction. It can be considered in the conclusions…

The methodology is satisfactory, but lack some information.

The information regarding the other antimicrobial therapies used is missing. This compromises the interpretation of the efficacy and security of MV, when compared with other therapies.

The number of elected studies to the analysis is too small and this fact should be considered as a main limitation of the present manuscript.

Why the authors excluded studies evolving pediatric patients? Could not be included as a third group?

The results and are correctly exposed, but need some clarifications….

Line 272: What were considered adverse events? Could the authors list, some of the main adverse events observed in both groups?

Lines 297 and 310: Could the authors clarify the differences in these two analyses? The criteria to those different analyses are not understandable.

The Discussion section is well structured and explains the results of the study. In my opinion, the adverse effects should be listed in more detail in the results section.

The limitations listed are real and reduces the impact of this analysis in the literature.

The conclusions are sound and cautious considering the result of the study.

Author Response

Dear Reviewer,

We appreciate the time and effort you dedicated to reviewing our work thoroughly. We have diligently addressed each point raised, implementing necessary corrections in our manuscript. We hope the revisions made using the tracked changes feature has significantly improved the quality of our manuscript.

The submitted manuscript “Systematic Review and Meta-analysis of Clinical Efficacy and Safety of Meropenem-Vaborbactam versus Best-Available Therapy in Patients with Carbapenem-resistant Enterobacteriaceae Infections” focus on an interesting and relevant subject regarding human health and antimicrobials resistance. Considering nowadays pertinence of the subject in the “one health” perspective, it is important to analyse scientific data and results to support the cautious use and efficacy of antimicrobials.

The manuscript aims to review and examine the efficacy and security of Meropenem-Vaborbactam versus other therapies. Even the number of patients of the different studies included in the analysis can be considered good, the number of studies elected in the Meta-analysis seems poor, maybe due to the scarce number of studies published in this subject or the exclusion criteria.

In general, the conceptualization of the study is correct.

However, the small number of studies elected to the analysis compromises the relevance of the present study and should be considered as a main limitation.

We agree with this and we have already included this main limitation of our analysis (lines 382-383).

This kind of scientific study is important and brings useful information to the literature.

The Abstract is satisfactory and well written. It provides a correct information regarding the content of the manuscript (objectives, materials and methods, results and discussion) in a clear and simple language.

Line: Can the authors specify better what consider BAT (best available therapy)? To the readers, BAT gives no useful information…

Thank you for your recommendation. We have updated the text in both the abstract and the main body of the manuscript. This change can be found at the lines 37-39 and 67-74, 84-88 (mono/combination therapy with polymyxins, carbapenems, aminoglycosides, colistin, tigecycline; or ceftazidime-avibactam; or piperacillin-tazobactam).

The keywords are adequate to the content of the study.

The introduction provides correct information regarding AMR and its importance, but can be improved.

Thank you for pointing this out. We agree with this comment. We have improved the introduction.

Lines 89-91: Can again clarify better what is considered the BAT (best available therapy). This concept is too unspecific to understand the usefulness of this manuscript.

We have explained better what was considered the BAT in the included studies. We have also modified Table S1 and lines 171-173.

Lines 91-92: This sentence seems too ambitious to be in the introduction. It can be considered in the conclusions…

We agree the sentence is too ambitious and we have rephrased it: „Our study results will contribute to informing future practices aimed at optimizing the treatment of CRE infections.”

The methodology is satisfactory, but lack some information.

The information regarding the other antimicrobial therapies used is missing. This compromises the interpretation of the efficacy and security of MV, when compared with other therapies.

The number of elected studies to the analysis is too small and this fact should be considered as a main limitation of the present manuscript.

We agree with this and we have already included this main limitation of our analysis (lines 384-385).

Why the authors excluded studies evolving pediatric patients? Could not be included as a third group?

We appreciate your insights and have deleted this exclusion creteria as we did not find any article envolving pediatric patients. This change can be found at the lines 129.

The results and are correctly exposed, but need some clarifications….

Line 272: What were considered adverse events? Could the authors list, some of the main adverse events observed in both groups?

Lines 297 and 310: Could the authors clarify the differences in these two analyses? The criteria to those different analyses are not understandable.

These analysis were performed by all four included studies and we have considered important as the authors considered also important. infections. Primary reasons

for clinical failure in both groups were death and discontinuation of study drug (due to either

death, clinical failure/need for additional antimicrobials, or AEs).The included studies presented the number of adverse events for both groups (M-V and BAT), respectively the number of study drug discontinuations due to treatment-emergent adverse event (TEAEs) and study discontinuations due to TEAEs.

The difference between "Discontinuation of study due to drug-related adverse events" and "Discontinuation of study drug due to drug-related adverse events" lies in what is being discontinued as a result of adverse events.

„Discontinuation of study drug due to drug-related adverse events” refers specifically to the discontinuation of the study drug or treatment regimen by participants due to adverse events. The participant stops taking the drug being tested but may still continue in the study for follow-up or other assessments.

„Discontinuation of study due to drug-related adverse events” refers to participants discontinuing their participation in the entire study due to adverse events. This means they are no longer involved in any aspect of the study, including follow-ups, assessments, or data collection. We have made changes in the manuscript to be clearer in the lines 323-325 and 338-340.

The Discussion section is well structured and explains the results of the study. In my opinion, the adverse effects should be listed in more detail in the results section.

We have added the adverse effects assessed in the included studies in the lines 307 and 319-321.

The limitations listed are real and reduces the impact of this analysis in the literature.

The conclusions are sound and cautious considering the result of the study.

Reviewer 2 Report

Comments and Suggestions for Authors

Dear authors,

The problem of carbapenem resistance is well described in the literature, and there have been many reviews and meta-analyses on this topic, clearly demonstrating and discussing treatment efficacy and clinical outcomes. M/V is well described, particularly in terms of efficacy against Klebsiella pneumoniae and other resistant Gram-negative bacilli.

Efficacy has recently been described here: 10.1007/s10096-024-04758-2, but the article does not fit the search key you used.

The review is valuable as knowledge of the clinical use of M/V is still limited.

However, please explain where such a high similarity coefficient of 41% comes from.

In addition, the aspect of Enterobacteriaceae resistance should be discussed more precious in the introduction and the most common CREs and their percentages should be given. Please also mention the bla KPC gene and other resistance genes bla NDM, bla OXA-48, bla IMP and bla VIM which are important for the background of the problem described.

In the introduction, please provide current CRE treatment guidelines that are relevant to the understanding of BAT.

Please explain why only 2 major databases were focused on when there are many more available including MEDLINE, Embase, LILACS, SciELO, CENTRAL, CINAHL, Cochrane Library, WHO Database.

In addition, whether the authors used words such as Vabomere OR (meropenem OR RPX-2014) OR (vaborbactam OR RPX-7009) when searching for information is crucial to the results obtained.

I have no detailed comments on the analysis itself. The conclusions are correct, as is the discussion. However, please indicate what the authors mean by BAT (which antibiotics were included in the review), this is important for the strategy of searching and analysing information, because I know of at least 7 old antibiotics and twice as many new ones used in the treatment of CRE. Please explain. Was it assumed that the searched articles would use the term BAT?

Given Table S1 and the drugs listed there (as BAT), why were the TANGO I and TANGO II trials not included in the review?

Although the literature review points to early administration within 48 hours as the key to reducing mortality. Many retrospective studies could be used in the discussion to extend and document the efficacy. I think it is worth widening the discussion.

Please explain, and after receiving an answer from them I will be able to recommend the article.

Kind Regards

REV.

Author Response

Dear Reviewer,

We appreciate the time and effort you dedicated to reviewing our work thoroughly. We have diligently addressed each point raised, implementing necessary corrections in our manuscript. We hope the revisions made using the tracked changes feature has significantly improved the quality of our manuscript.

The problem of carbapenem resistance is well described in the literature, and there have been many reviews and meta-analyses on this topic, clearly demonstrating and discussing treatment efficacy and clinical outcomes. M/V is well described, particularly in terms of efficacy against Klebsiella pneumoniae and other resistant Gram-negative bacilli.

Efficacy has recently been described here: 10.1007/s10096-024-04758-2, but the article does not fit the search key you used.

We have added this article in our discussion section.

The review is valuable as knowledge of the clinical use of M/V is still limited.

However, please explain where such a high similarity coefficient of 41% comes from.

We appologize, but we do not find this outcome to explain.

In addition, the aspect of Enterobacteriaceae resistance should be discussed more precious in the introduction and the most common CREs and their percentages should be given. Please also mention the bla KPC gene and other resistance genes bla NDM, bla OXA-48, bla IMP and bla VIM which are important for the background of the problem described.

Agree. We have accordingly revised the introduction to empasize this point.

This change can be found at the lines 70-74.

In the introduction, please provide current CRE treatment guidelines that are relevant to the understanding of BAT.

This change can be found at lines 84-88.

Please explain why only 2 major databases were focused on when there are many more available including MEDLINE, Embase, LILACS, SciELO, CENTRAL, CINAHL, Cochrane Library, WHO Database.

We agree there are many more available databases. We have focused on PubMed and Scopus because they are freely accessible databases primarily focused on healthcare. PubMed contains references and abstracts from MEDLINE, and every article indexed in MEDLINE is also indexed in PubMed (MEDLINE is part of PubMed). Scopus indexes over 22,000 journals, more than Embase (indexes over 8,500 journals). Additionally, Embase is not a free-to-access platform and we can’t access Embase through our university. The strengths of LILACS (Latin American and Caribbean Health Sciences Literature) and SciELO are that provide access to literature in Spanish, Portuguese, and other languages relevant to Latin America and the Caribbean. We have focused on articles in English.

In addition, whether the authors used words such as Vabomere OR (meropenem OR RPX-2014) OR (vaborbactam OR RPX-7009) when searching for information is crucial to the results obtained.

We have used the proposed words Vabomere OR ((meropenem OR RPX-2014) AND (vaborbactam OR RPX-7009)), but we didn’t find more articles. Thank you for pointing this out. We have changed the manuscript in lines 121-122.

I have no detailed comments on the analysis itself. The conclusions are correct, as is the discussion. However, please indicate what the authors mean by BAT (which antibiotics were included in the review), this is important for the strategy of searching and analysing information, because I know of at least 7 old antibiotics and twice as many new ones used in the treatment of CRE. Please explain. Was it assumed that the searched articles would use the term BAT?

We have explained better what was considered the BAT in the included studies. We have added more information in the abstract (lines 37-39) and in the introduction (84-88). We have also modified Table S1 and added more information in lines 171-173.

Given Table S1 and the drugs listed there (as BAT), why were the TANGO I and TANGO II trials not included in the review?

TANGO I and TANGO II trials was included as Kaye et al. 2018 and Wunderink et al. 2018, respectively.

Although the literature review points to early administration within 48 hours as the key to reducing mortality. Many retrospective studies could be used in the discussion to extend and document the efficacy. I think it is worth widening the discussion.

Thank you for pointing this out. We agree with this comment. We have added more details at the discussion. This change can be found at the lines 373-377.

Round 2

Reviewer 2 Report

Comments and Suggestions for Authors

Dear authors,

Thank you for completing the paper and addressing my concerns.

I accept the paper as it stands.

Kind Regards

REV.